# Fixed point results for ℑ-Contractions in JS-generalized metric spaces with an application

**Bilal Iqbal[1]☺, Naeem Saleem☺[1,2]☺\*, Maggie Aphane[2]☺, Asima Razzaque[3,4]☺**

**1** Department of Mathematics, University of Management and Technology, Lahore, Pakistan, **2** Department of Mathematics and Applied Mathematics, Sefako Makgatho Health Sciences University, Ga-Rankuwa, Pretoria, South Africa, **3** Department of Basic Sciences, Preparatory Year, King Faisal University, Al-Ahsa, Saudi Arabia, **4** Department of Mathematics, College of Science, King Faisal University, Al-Ahsa, Saudi Arabia

☺ These authors contributed equally to this work.
\* naeem.saleem2@gmail.com

## Abstract

The goal of this work is to establish ℑ-contractions and to show some novel fixed point theorems for these contractive conditions in the setting of generalized metric spaces in the sense of Jleli and Samet. Finally, using proven fixed point results, an existence result for a solution of the RLC circuit's current differential equation is established.

**Data Availability Statement:** We confirm that all information needed to replicate our study in its entirety has been provided in paper.

**Funding:** This work was supported by the Deanship of Scientific Research, Vice Presidency

## 1 Introduction

The Banach fixed point theorem [1] served as the inspiration for metric fixed point theory. Because this approach has many applications in several disciplines, many authors have expanded it in many ways [2–5]. Wardowski [6] provides one such astonishing and significant generalization. He introduced the notion of ℑ-contraction as follows:

**Definition 0.1**. *Let $(z, m)$ be a metric space. A mapping $\xi: z \to z$ is said to be an ℑ-contraction, if there is $\mathfrak{I} \in \Delta(\mathfrak{I})$ and $\lambda > 0$ such that for all $x, y \in z$*

$$\lambda + \mathfrak{I}(m(\xi x, \xi y)) \leq \mathfrak{I}(m(x, y)), \tag{1}$$

*where $\Delta(\mathfrak{I})$ is the family of all mappings $\mathfrak{I} : (0, +\infty) \to \mathbb{R}$ obeying the following conditions:*

$(\mathfrak{I}_1)$ $\mathfrak{I}(x) < \mathfrak{I}(y)$ *for all* $x < y$;

$(\mathfrak{I}_2)$ *for all sequences* $\{\eta_p\} \subseteq (0, +\infty)$, $\lim_{p \to \infty} \eta_p = 0$, *if and only if* $\lim_{p \to \infty} \mathfrak{I}(\eta_p) = -\infty$;

$(\mathfrak{I}_3)$ *there exists* $0 < \ell < 1$ *such that* $\lim_{\eta \to 0^+} \eta^\ell \mathfrak{I}(\eta) = 0$.

Wardowski's result is given as follows:

**THEOREM 0.1** [6]. *Let $(z, m)$ be a complete metric space and $\xi: z \to z$ be ℑ-contraction. Then $x^* \in z$ is a unique fixed point of $\xi$ and for each $x_0 \in z$ the sequence $\{\xi^p x_0\}_{p \in \mathbb{N}}$ is convergent to $x^*$.*

In [7], Secelean demonstrated that condition $(\mathfrak{I}_2)$ may be overtaken by an equivalent and simpler one.

for Graduate Studies and Scientific Research, King Faisal University, Saudi Arabia [KFU242460].

**Competing interests:** The authors have declared that no competing interests exist.

$(\Im'_2) \lim_{t \to 0^+} \Im(t) = -\infty$.

**Lemma 0.1.** *Let* $\Im : (0, \infty) \to \mathbb{R}$ *be a function obeying* $(\Im_1)$ *and* $(\Im'_2)$, *then for all sequence* $\{t_p\} \subseteq (0, \infty)$

$$\Im(t_p) \to -\infty \quad implies \quad t_p \to 0.$$

Following that, Piri and Kumam [8] established Wardowski's theorem utilising $(\Im'_2)$ and the continuity of $\Im$ rather than $(\Im_2)$ and $(\Im_3)$, respectively. Wardowski [9] later proved a fixed point theorem for $\Im$-contractions when $\lambda$ is treated as a function. Recently, other authors demonstrated (in various methods) Wardowski's original results in the absence of both criterias $(\Im_2)$ and $(\Im_3)$ (see, [10, 11]. To more in this direction, consult [12–15]. Very recently, Derouiche and Ramoul [16] used a relaxed version of $(\Im_2)$ and also dropped $(\Im_3)$ to establish some new fixed point results in the context of b-metric spaces.

On the other side, the concept of standard metric space is generalized in numerous ways (see [17–21]. Jleli and Samet [22] recapitulated a huge class of topological spaces by introducing the most prevailing generalizations of metric spaces namely JS-generalized metric spaces. More far, in [23], Karapinar *et al.* achieved fixed point theorems under very general contractive conditions and Altun *et al.* [24] proved a Feng-Liu's type fixed point theorem in JS-generalized metric spaces. While, in [25], Dumitrescu and Pitea presented extensions of some classic results regarding the existence and uniqueness of fixed points of operators fulfilling generalized contractive conditions in the setting of JS-generalized metric spaces. Quite recently, Saleem *et al.* [26] proved some new fixed point theorems, coincidence point theorems and common fixed point theorems for multivalued $\Im$-contractions in the framework of JS-generalized metric spaces. Afterwards, Iqbal *et al.* [27] derived the coincidence point and common fixed point results for $\Im$-type mappings with regard to JS-generalized metric spaces.

A binary relation on $z$ is a non-empty subset $\sim_{\mathcal{R}}$ of the Cartesian product $z \times z$. For ease of use, we designate $x \sim_{\mathcal{R}} y$ if $(x, y) \in \sim_{\mathcal{R}}$. The concepts of antisymmetry, preorder, reflexivity, transitivity, and partial order can be found in [28]. The trivial preorder on $z$ is denoted by $\sim_{\mathcal{R}_z}$, and is given by $x \sim_{\mathcal{R}_z} y$ for each $x, y \in z$. Here after, $\mathbb{R}$ and $\mathbb{N}$ demonstrate the set of real numbers and the set of non-negative integers, respectively. Let $z$ be a non-empty set and $J_d: z \times z \to [0, \infty]$ be a given mapping. Following Jleli and Samet [22], for every $x \in z$, define the set

$$c(J_d, z, x) = \left\{ \{x_p\} \subset z : \lim_{p \to \infty} J_d(x_p, x) = 0 \right\}. \tag{2}$$

**Definition 0.2** [22]. *Let* $z$ *be a non-empty set and* $J_d: z \times z \to [0, \infty]$ *be a function that complies with the following criteria for all* $x, y \in z$:

$(J_{d1})$ $J_d(x, y) = 0$ *implies* $x = y$;

$(J_{d2})$ $J_d(x, y) = J_d(y, x)$;

$(J_{d3})$ *there exist* $\kappa > 0$ *such that* $(x, y) \in z \times z, \{x_p\} \in J_d, z, x)$ *implies*

$$J_d(x, y) \leq \kappa \lim_{p \to \infty} \sup J_d(x_p, y). \tag{3}$$

*Then* $J_d$ *is called a JS-generalized metric and the pair* $(z, J_d)$ *is called a JS-generalized metric space. We renamed it as* $\kappa$-*JS-generalized metric space (in short, a* $\kappa_{\mathcal{JS}}$-*MS).*

**Remark 0.1** [22]. *If the set* $c(J_d, z, x)$ *is empty for every* $x \in z$, *then* $(z, J_d)$ *is a* $\kappa_{\mathcal{JS}}$-*MS if and only if* $(J_{d1})$ *and* $(J_{d2})$ *are satisfied.*

Many examples of $\kappa_{\mathcal{JS}}$-MS can be found in [22, 23, 26].

**Example 0.1** [22].

1. *A modular metric space $(z, \rho)$ is a $\rho_{\mathcal{JS}}$-MS.*

2. *A standard metric space is a $1_{\mathcal{JS}}$-MS.*

3. *A 2-metric space is a $2_{\mathcal{JS}}$-MS.*

**Definition 0.3**. [22] *Let $(z, J_d)$ be a $\kappa_{\mathcal{JS}}$-MS and $\mathrm{x} \in z$.*

1. *A sequence $\{x_p\} \subseteq z$ is said to be $J_d$-convergent and $J_d$-converges to $\mathrm{x}$ if $\{x_p\} \in J_d, z, \mathrm{x})$. In such case, we will write $\{x_p\} \xrightarrow{J_d} \mathrm{x}$.*

2. *a sequence $\{x_p\} \subseteq z$ is said to be $J_d$-Cauchy if*

$$\lim_{p,q \to \infty} J_d(x_p, x_{p+q}) = 0. \tag{4}$$

3. *A $\kappa_{\mathcal{JS}}$-MS $(z, J_d)$ is said to be complete if every $J_d$-Cauchy sequence in $z$ is $J_d$-convergent.*

**Proposition 0.1** [22]. *Let $(z, J_d)$ be a $\kappa_{\mathcal{JS}}$-MS, $\{x_p\}$ be a sequence in $z$ and $(\mathrm{x}, \mathrm{y}) \in z \times z$. If $\{x_p\}$ is $J_d$-convergent and $J_d$-converges to $\mathrm{x}$ and $\{x_p\}$ $J_d$-converges to $\mathrm{y}$, then $\mathrm{x} = \mathrm{y}$.*

**Remark 0.2**. *Jleli and Samet defined $J_d$-Cauchy sequence as*

$$\lim_{p,q \to \infty} J_d(x_p, x_q) = 0. \tag{5}$$

*Clearly, (5) implies (4), the opposite, however, need not be true [23]. From here on, we assume that $J_d$-Cauchy sequences are given by (5).*

**Definition 0.4** [23]. *Let $(z, J_d)$ be a $\kappa_{\mathcal{JS}}$-MS and $\xi: z \to z$. For $x_0 \in z$, denote $\delta(J_d, \xi, x_0)$, the $J_d$-diameter of the orbit of $x_0$ by $\xi$, $\mathcal{O}_\xi(x_0) = \{\xi^p x_0 : p \in \mathbb{N}\}$, and is defined as,*

$$\delta(J_d, \xi, x_0) = \sup(\{J_d(\xi^p x_0, \xi^q x_0) : q, p \in \mathbb{N}\}). \tag{6}$$

**Definition 0.5** [23]. *Let $\sim_{\mathcal{R}}$ be a binary relation on $\kappa_{\mathcal{JS}}$-MS $(z, J_d)$. A sequence $\{x_p\} \subseteq z$ is $\sim_{\mathcal{R}}$-non-decreasing if $x_p \sim_{\mathcal{R}} x_{p+1}$ for each $p \in \mathbb{N}$.*

**Definition 0.6** [23]. *A $\kappa_{\mathcal{JS}}$-MS $(z, J_d)$ is called $\sim_{\mathcal{R}}$-non-decreasing complete if every $\sim_{\mathcal{R}}$-non-decreasing and $J_d$-Cauchy sequence is $J_d$-convergent in $z$.*

**Remark 0.3**. *Notice that every complete $\kappa_{\mathcal{JS}}$-MS is also $\sim_{\mathcal{R}}$-non-decreasing complete. As evidenced by the case below, the contrary is untrue.*

**Example 0.2**. *Let $z = (0, 1]$ furnished with the Euclidean metric $m(\mathrm{x}, \mathrm{y}) = |\mathrm{x} - \mathrm{y}|$ for each $\mathrm{x}, \mathrm{y} \in z$. Define a binary relation $\sim_{\mathcal{R}}$ on $z$ by*

$$\mathrm{x} \sim_{\mathcal{R}} \mathrm{y} \quad if \quad 0 < \mathrm{x} \le \mathrm{y} \le 1.$$

*Then $(z, m)$ is $\sim_{\mathcal{R}}$-non-decreasing complete, however, the metric space is not complete.*

**Definition 0.7** [23]. *Let $(z, J_d)$ be a $\kappa_{\mathcal{JS}}$-MS. A mapping $\xi: z \to z$ is $\sim_{\mathcal{R}}$-non-decreasing-continuous at $v \in z$ if $\{\xi x_p\} \in J_d, z, sv)$ for each $\sim_{\mathcal{R}}$-non-decreasing sequence $\{x_p\} \in J_d, z, v)$. A mapping $\xi$ is $\sim_{\mathcal{R}}$-non-decreasing-continuous if it is $\sim_{\mathcal{R}}$-non-decreasing-continuous at every point of $z$.*

**Remark 0.4**. [23] *Every continuous mapping is also $\sim_{\mathcal{R}}$-non-decreasing-continuous, while the reverse is generally false, as seen in Example 4.6 of [23].*

By getting inspiration from the work of Karapinar *et al.* [23], here, we prove fixed point theorems for ℑ-contractions in the context of JS-generalized metric space.

## 2 Fundamental results

Let $(z, J_d)$ be a $\kappa_{\mathcal{JS}}$-MS and let $\xi$ be a self-mapping on $z$. Throughout this section, we denote, for all $x, y \in z$,

$$\mathcal{Q}_{\xi}^{J_d}(x, y) = \max\{J_d(x, y), J_d(x, \xi x), J_d(y, \xi y), J_d(x, \xi y), J_d(y, \xi x)\}.$$

Following [23], define for given $p_0 \in \mathbb{N}$,

$$\delta_{p_0}(J_d, \xi, x_0) = \sup(\{J_d(\xi^p x_0, \xi^q x_0) : q, p \in \mathbb{N}, q, p \geq p_0\}). \tag{7}$$

and

$$\delta(J_d, \xi, x_0) = \delta_0(J_d, \xi, x_0) = \sup(\{J_d(\xi^p x_0, \xi^q x_0) : q, p \in \mathbb{N}\}). \tag{8}$$

By the symmetry of $J_d$, we can alternatively express

$$\delta_{p_0}(J_d, \xi, x_0) = \sup(\{J_d(\xi^p x_0, \xi^q x_0) : q, p \in \mathbb{N}, q \geq p \geq p_0\}). \tag{9}$$

Notice that if $q, p \in \mathbb{N}$ satisfy $q \geq p$, then

$$\delta_q(J_d, \xi, x_0) \leq \delta_p(J_d, \xi, x_0) \leq \delta(J_d, \xi, x_0). \tag{10}$$

**Lemma 1.1**. *Let $(z, J_d)$ be a $\kappa_{\mathcal{JS}}$-MS with a preorder $\sim_{\mathcal{R}}$ and let $\xi: z \to z$ be an $\sim_{\mathcal{R}}$ -non-decreasing mapping. Let $x_0 \in z$ be a point such that $x_0$ and $\xi x_0$ are $\sim_{\mathcal{R}}$ -comparable. Assume, there is a function $\mathfrak{I} : (0, \infty) \to \mathbb{R}$ such that*

$$\xi x \neq \xi y \quad \text{implies} \quad \lambda + \mathfrak{I}(J_d(\xi x, \xi y)) \leq \mathfrak{I}(\mathcal{Q}_{\xi}^{J_d}(x, y)), \tag{11}$$

*for $x, y \in z$ satisfying $x \sim_{\mathcal{R}} y$ and $\lambda > 0$. Then (11) holds for each $x, y \in \mathcal{O}_{\xi}(x_0)$.*

*Proof.* Consider the Picard sequence $\{x_p = \xi x_{p-1} = \xi^p x_0\}_{p \in \mathbb{N}}$ of $\xi$ based on $x_0$. Suppose that $x_0 \sim_{\mathcal{R}} x_1$. As $\xi$ is $\sim_{\mathcal{R}}$ -non-decreasing, then $x_1 = \xi x_0 \sim_{\mathcal{R}} \xi x_1 = x_2$. Repeating this argument, we get, $x_p \sim_{\mathcal{R}} x_{p+1}$ for every $p \in \mathbb{N}$. Since $\sim_{\mathcal{R}}$ is a preorder, then $x_p \sim_{\mathcal{R}} x_q$ for all $q, p \in \mathbb{N}$ such that $p \leq q$. Furthermore, as condition (11) is symmetric on $x$ and $y$, then (11) holds for each $x_p$ and $x_q$ (being $q, p \in \mathbb{N}$ arbitrary), so it holds for each $x, y \in \mathcal{O}_{\xi}(x_0)$.

**Lemma 1.2**. *Let $(z, J_d)$ be a $\kappa_{\mathcal{JS}}$-MS and let $\xi: z \to z$ be a mapping. Let $x_0 \in z$ be a point for which there exists $p_0 \in \mathbb{N}$ such that $\delta_{p_0}(J_d, \xi, x) < \infty$. Assume, there is a non-decreasing function $\mathfrak{I} : (0, \infty) \to \mathbb{R}$ obeying*

$(\mathfrak{I}_{J_d})$: $\mathfrak{I}(\sup M) = \sup \mathfrak{I}(M)$ *for all $M \subset (0, \infty)$ with $\sup M > 0$*

   *and*

$$\xi x \neq \xi y \quad \text{implies} \quad \lambda + \mathfrak{I}(J_d(\xi x, \xi y)) \leq \mathfrak{I}(\mathcal{Q}_{\xi}^{J_d}(x, y)), \tag{12}$$

*for all $x, y \in \mathcal{O}_{\xi}(x_0)$ and $\lambda > 0$. Then*

$$\delta_{\ell+1}(J_d, \xi, x_0) \leq \delta_{\ell}(J_d, \xi, x_0) - \lambda, \quad \text{for all} \quad \ell \in \mathbb{N}, \ell \geq p_0.$$

*In particular,*

$$\delta_{p_0+\ell}(J_d, \xi, x_0) \leq \delta_{p_0}(J_d, \xi, x_0) - \ell\lambda, \quad \text{for all} \quad \ell \in \mathbb{N}, \ell \geq p_0.$$

*Proof.* Let $\ell \in \mathbb{N}$ be such that $\ell \geq p_0$. From (10), we have

$$\delta_{\ell+1}(J_d, \xi, x_0) \leq \delta_{\ell}(J_d, \xi, x_0) \leq \delta_{p_0}(J_d, \xi, x_0) < \infty.$$

Let $q, p \in \mathbb{N}$ be such that $p \geq q \geq \ell + 1$. Denote

$$\Omega_\ell = \{J_d(\xi^r x_0, \xi^s x_0) : r, s \in \mathbb{N}, \quad r, s \geq \ell\},$$

then

$$J_d(\xi^{p-1} x_0, \xi^{q-1} x_0), J_d(\xi^{p-1} x_0, \xi^p x_0), J_d(\xi^{q-1} x_0, \xi^q x_0), J_d(\xi^{p-1} x_0, \xi^q x_0), J_d(\xi^{q-1} x_0, \xi^p x_0) \in \Omega_\ell.$$

Hence

$$\begin{aligned}
&\max\{J_d(\xi^{p-1} x_0, \xi^{q-1} x_0), J_d(\xi^{p-1} x_0, \xi^p x_0), J_d(\xi^{q-1} x_0, \xi^q x_0), J_d(\xi^{p-1} x_0, \xi^q x_0), \\
&J_d(\xi^{q-1} x_0, \xi^p x_0)\} \leq \sup\Omega_\ell = \delta_\ell(J_d, \xi, x_0).
\end{aligned} \tag{13}$$

From (13) and (13), we obtain

$$\begin{aligned}
\lambda + \mathfrak{I}(J_d(\xi^p x_0, \xi^q x_0)) &\leq \mathfrak{I}(\mathcal{Q}_\xi^{J_d}(\xi^{p-1} x_0, \xi^{q-1} x_0)) \\
&\leq \mathfrak{I}(\delta_\ell(J_d, \xi, x_0)).
\end{aligned} \tag{14}$$

By the virtue of $(\mathfrak{I}_{J_d})$ and (14), we get

$$\begin{aligned}
\mathfrak{I}(\delta_{\ell+1}(J_d, \xi, x_0)) &= \mathfrak{I}(\sup\{J_d(\xi^p x_0, \xi^q x_0) : q, p \in \mathbb{N}, \ \ell + 1 \leq n \leq p\}) \\
&= \sup\{\mathfrak{I}(J_d(\xi^p x_0, \xi^q x_0)) : q, p \in \mathbb{N}, \ \ell + 1 \leq n \leq p\} \\
&\leq \sup\{\mathfrak{I}(\delta_\ell(J_d, \xi, x_0)) - \lambda : q, p \in \mathbb{N}, \ \ell + 1 \leq n \leq p\} \\
&\leq \mathfrak{I}(\delta_\ell(J_d, \xi, x_0)) - \lambda.
\end{aligned} \tag{15}$$

Continuing this argument and recognizing that $\mathfrak{I}$ is non-decreasing, we get for all $\ell \in \mathbb{N}$

$$\begin{aligned}
\mathfrak{I}(\delta_{p_0+\ell}(J_d, \xi, x_0)) &\leq \mathfrak{I}(\delta_{p_0+\ell-1}(J_d, \xi, x_0)) - \lambda \\
&\leq \mathfrak{I}(\delta_{p_0+\ell-2}(J_d, \xi, x_0)) - 2\lambda \\
&\vdots \\
&\leq \mathfrak{I}(\delta_{p_0}(J_d, \xi, x_0)) - \ell\lambda.
\end{aligned}$$

# 3 Main results

**THEOREM 2.1.** *Let $(z, J_d)$ be a $\kappa_{\mathcal{JS}}$-MS with a preorder $\sim_{\mathcal{R}}$ and let $\xi: z \to z$ be an $\sim_{\mathcal{R}}$-non-decreasing mapping. Let $x_0 \in z$ be a point such that $x_0 \sim_{\mathcal{R}} \xi x_0$ and $\delta_{p_0}(J_d, \xi, x) < \infty$ for some $p_0 \in \mathbb{N}$. Assume, there is a non-decreasing function $\mathfrak{I} : (0, \infty) \to \mathbb{R}$ obeying $(\mathfrak{I}'_2), (\mathfrak{I}_{J_d})$ and (11) for all $x, y \in \mathcal{O}_\xi(x_0)$ and $\lambda > 0$. Then the sequence $\{\xi^p x_0\}_{p \in \mathbb{N}}$ based on $x_0$ is $\sim_{\mathcal{R}}$-non-decreasing and $J_d$-Cauchy sequence.*

*Furthermore, if $(z, J_d)$ is $\sim_{\mathcal{R}}$-non-decreasing-complete, then $\{\xi^p x_0\}_{p \in \mathbb{N}}$ $J_d$-converges to a point $v \in z$ that obeys*

$$J_d(v, v) = 0.$$

*Additionally, if $\xi$ is $\sim_{\mathcal{R}}$-non-decreasing-continuous, then $v = \xi v$.*

*Proof.* Consider the Picard sequence $\{x_p = \xi x_{p-1} = \xi^p x_0\}_{p \in \mathbb{N}}$ of $\xi$ based on $x_0$. As shown in the proof of Lemma 1.1, $\{\xi^p x_0\}_{p \in \mathbb{N}}$ is $\sim_{\mathcal{R}}$-non-decreasing. If $\xi^p x_0 = \xi^q x_0$, then $J_d(x_p, x_q) = 0$ for

every q, p $\geq$ $p_0$. In particular,

$$\lim_{p,q\to\infty} J_d(x_p, x_q) = 0.$$

Consider $\xi^p x_0 \neq \xi^q x_0$ and q, p $\in \mathbb{N}$, $p_0 \leq p < q$ for some $p_0 \in \mathbb{N}$, then by using Lemma 1.2, we have

$$\mathfrak{I}(J_d(\xi^p x_0, \xi^q x_0)) \leq \mathfrak{I}(\delta_{q+p}(J_d, \xi, x_0)) \leq \mathfrak{I}(\delta_q(J_d, \xi, x_0)) - p\lambda. \tag{16}$$

Letting limit in (16) as p, q $\to \infty$, we have

$$\lim_{p,q\to\infty} \mathfrak{I}(J_d(\xi^p x_0, \xi^q x_0)) = \infty. \tag{17}$$

Taking into account of $(\mathfrak{I}_2')$, we have

$$\lim_{p,q\to\infty} J_d(\xi^p x_0, \xi^q x_0) = 0.$$

Hence $\{\xi^p x_0\}_{p\in\mathbb{N}}$ $J_d$-Cauchy sequence. Since $(z, J_d)$ is $\sim_{\mathcal{R}}$ -non-decreasing-complete, there exists $v \in z$ such that $\{x_p\} \xrightarrow{J_d} v$. By using $(J_{d3})$, we get

$$0 \leq J_d(v, v) \leq \kappa \limsup_{p\to\infty} J_d(x_p, v) = 0, \tag{18}$$

which implies $J_d(v, v) = 0$.

Moreover, as we additionally assume that $\xi$ is $\sim_{\mathcal{R}}$ -non-decreasing-continuous, then

$$\{x_{p+1} = \xi^p x_0 = \xi x_p\} \xrightarrow{J_d} \xi v.$$

Proposition 0.1 guarantees that $\xi v = v$, so $v$ is a fixed point of $\xi$.

**Example 2.1**. *Consider the function* $\mathfrak{I} : (0, \infty) \to \mathbb{R}$ *defined as*

$$\mathfrak{I}(t) = t^t \quad for \ all \quad t \in (0, \infty).$$

*Then* $\mathfrak{I}$ *is non-decreasing and continuous but does not satisfies* $(\mathfrak{I}_2')$.

So, in next theorem, we replace the condition of $(\mathfrak{I}_2')$ by the continuity of $\mathfrak{I}$ in Theorem 2.1 and we denote by $\mathcal{W}$, the collection of all functions $\mathfrak{I} : (0, \infty) \to \mathbb{R}$ that are continuous and non-decreasing.

**THEOREM 2.2**. *Let* $(z, J_d)$ *be a* $\kappa_{\mathcal{JS}}$-*MS with a preorder* $\sim_{\mathcal{R}}$ *and let* $\xi$: $z \to z$ *be an* $\sim_{\mathcal{R}}$ -*non-decreasing self-mapping. Let* $x_0 \in z$ *be a point such that* $x_0 \sim_{\mathcal{R}} \xi x_0$, $\delta_{p_0}(J_d, \xi, x) < \infty$ *for some* $p_0 \in \mathbb{N}$ *and the following holds true*:

$$\lim_{p_0+\ell\to\infty} \delta_{p_0+\ell}(J_d, \xi, x_0) = \lim_{p_0\to\infty} \delta_{p_0}(J_d, \xi, x_0) \quad for \ arbitrary \quad \ell \in \mathbb{N} \quad such \ that \quad \ell \geq p_0.$$

*If there exists a function* $\mathfrak{I} \in \mathcal{W}$ *and* $\lambda > 0$ *such that inequality* (11) *holds for all* x, y $\in \mathcal{O}_\xi(x_0)$, *then the sequence* $\{\xi^p x_0\}_{p\in\mathbb{N}}$ *based on* $x_0$ *is* $\sim_{\mathcal{R}}$ -*non-decreasing and* $J_d$-*Cauchy sequence.*

*Furthermore, if* $(z, J_d)$ *is* $\sim_{\mathcal{R}}$ -*non-decreasing-complete, then* $\{\xi^p x_0\}_{p\in\mathbb{N}}$ $J_d$-*converges to a point* $v \in z$ *that satisfies*

$$J_d(v, v) = 0.$$

*Additionally, if* $\xi$ *is* $\sim_{\mathcal{R}}$ -*non-decreasing-continuous, then* $v = \xi v$.

*Proof.* Consider the Picard sequence $\{x_p = \xi x_{p-1} = \xi^p x_0\}_{p\in\mathbb{N}}$ of $\xi$ based on $x_0$. As shown in the proof of Lemma 1.1, $\{\xi^p x_0\}_{p\in\mathbb{N}}$ is $\sim_{\mathcal{R}}$ -non-decreasing. If $\delta_{p_0}(J_d, \xi, x_0) = 0$, then $J_d(x_p, x_q) =$

0 for all q, p $\geq$ p$_0$. In particular,

$$\lim_{q,p\to\infty} J_d(x_p, x_q) = 0.$$

Consider $\delta_{p_0}(J_d, \xi, x_0) > 0$ and $q, p \in \mathbb{N}, p_0 \leq p < q$ for some $p_0 \in \mathbb{N}$. Denote

$$\Omega_{p_0} = \{J_d(\xi^r x_0, \xi^s x_0) : r, s \in \mathbb{N}, \quad r, s \geq p_0\},$$

then

$$J_d(\xi^{p-1} x_0, \xi^{q-1} x_0), J_d(\xi^{p-1} x_0, \xi^p x_0), J_d(\xi^{q-1} x_0, \xi^q x_0), J_d(\xi^{p-1} x_0, \xi^q x_0), J_d(\xi^{q-1} x_0, \xi^p x_0) \in \Omega_{p_0}.$$

Hence

$$\begin{aligned}
&\max\{J_d(\xi^{p-1} x_0, \xi^{q-1} x_0), J_d(\xi^{p-1} x_0, \xi^p x_0), J_d(\xi^{q-1} x_0, \xi^q x_0), J_d(\xi^{p-1} x_0, \xi^q x_0), \\
&J_d(\xi^{q-1} x_0, \xi^p x_0)\} \leq \sup \Omega_{p_0} = \delta_{p_0}(J_d, \xi, x_0).
\end{aligned} \tag{19}$$

Assume that $\lim_{p_0+\ell\to\infty} \delta_{p_0+\ell}(J_d, \xi, x_0) = \lim_{p_0\to\infty} \delta_{p_0}(J_d, \xi, x_0) = \mho$ for $\ell \in \mathbb{N}$ be an arbitrary such that $\ell \geq p_0$. Then, by using inequalities (11) and (19) and continuity of $\mathfrak{I}$, we obtain

$$\begin{aligned}
\lambda + \mathfrak{I}(\mho) &= \lambda + \mathfrak{I}\left(\lim_{p_0+\ell\to\infty} \delta_{p_0+\ell}(J_d, \xi, x_0)\right) \\
&= \lambda + \mathfrak{I}\left(\limsup_{p,q\to\infty} J_d(\xi^p x_0, \xi^q x_0)\right) \\
&= \limsup_{p,q\to\infty} \left(\lambda + \mathfrak{I}(J_d(\xi^p x_0, \xi^q x_0))\right) \\
&\leq \limsup_{p,q\to\infty} \left(\mathfrak{I}(\delta_{p_0}(J_d, \xi, x_0))\right) \\
&= \mathfrak{I}\left(\lim_{p_0\to\infty} \delta_{p_0}(J_d, \xi, x_0)\right) \\
&= \mathfrak{I}(\mho),
\end{aligned} \tag{20}$$

a contradiction because $\lambda > 0$. Hence $\{\xi^p x_0\}_{p\in\mathbb{N}}$ $J_d$-Cauchy sequence. Since $(z, J_d)$ is $\sim_{\mathcal{R}}$ -non-decreasing-complete, there is $v \in z$ such that $\{x_p\} \xrightarrow{J_d} v$. By using ($J_{d3}$), we get

$$0 \leq J_d(v, v) \leq \kappa \limsup_{p\to\infty} J_d(x_p, v) = 0, \tag{21}$$

which implies $J_d(v, v) = 0$.

Moreover, as we additionally assume that $\xi$ is $\sim_{\mathcal{R}}$ -non-decreasing-continuous, then

$$\{x_{p+1} = \xi^p x_0 = \xi x_p\} \xrightarrow{J_d} \xi v.$$

Proposition 0.1 guarantees that $\xi v = v$.

**Example 2.2**. *Let $z = [0, 1] \cup \{2\}$ and let $J_d: z \times z \to [0, \infty]$ be a function defined by*

$$J_d(x, y) = \begin{cases} 10 & \text{if either} \quad (x, y) = (0, 2) \quad (x, y) = (2, 0), \\ |x - y| & \text{otherwise.} \end{cases}$$

*Then $(z, J_d)$ is complete $\kappa_{\mathcal{JS}}$-MS (see [23]. Define a binary relation $\sim_{\mathcal{R}}$ on $z$ by*

$$x \sim_{\mathcal{R}} y \quad \text{if} \quad 0 < x \leq y \leq 1,$$

*then $\sim_{\mathcal{R}}$ is a preorder and $(z, J_d)$ is a preordered space. Let $x_0 = 0.25 \in z$ be a point such that*

$0 < x_0 = 0.25 < 1 = \xi(0.25) = \xi x_0$, *then* $x_0 \sim_{\mathcal{R}} \xi x_0$ *and*

$$\mathcal{O}_\xi(0.25) = \{0.25, 1, 0.5, 0.5, 0.5, \cdots\}.$$

*Also,* $\delta_{p_0}(J_d, \xi, x) = 0.5 < \infty$ *and* $\lim_{p_0+\ell \to \infty} \delta_{p_0+\ell}(J_d, \xi, x_0) = \lim_{p_0 \to \infty} \delta_{p_0}(J_d, \xi, x_0)$ *for any* $p_0, \ell \in \mathbb{N}$ *such that* $\ell \geq p_0$. *Now Define* $\xi: z \to z$ *and* $\Im: (0, \infty) \to (-\infty, \infty)$ *by*

$$\xi(x) = \begin{cases} 0.5 & \text{if} \quad x \in \{0.5, 1\}, \\ 1 & \text{if} \quad x = 0.25, \\ 0 & \text{otherwise} \end{cases} \quad \text{and} \quad \Im(r) = \ln r \quad \text{for all} \quad r \in (0, \infty)$$

*respectively, then* $\Im$ *is continuous and non-decreasing function. Let* $x, y \in \mathcal{O}_\xi(0.25)$ *such that* $\xi x \neq \xi y$, *then there arises two cases*:

**Case 1:** *When* $x = 0.25$, $y = 0.5$, *then there exists* $\lambda = 0.25 > 0$ *such that*

$$\Im(J_d(\xi x, \xi y)) - \Im(\mathcal{Q}_\xi^{J_d}(x, y))$$
$$= \Im(J_d(\xi 0.25, \xi 0.5)) - \Im(\mathcal{Q}_\xi^{J_d}(0.25, 0.5))$$
$$= \Im(J_d(1, 0.5)) - \Im(\max\{J_d(0.25, 0.5), J_d(0.25, 1), J_d(0.5, 0.5), J_d(0.25, 0.5), J_d(0.5, 1)\})$$
$$= \Im(0.5) - \Im(\max\{0.25, 0.75, 0, 0.25, 0.5\})$$
$$= \Im(0.5) - \Im(0.75) = \ln(0.5) - \ln(0.75)$$
$$= -0.4055 < -0.25 = -\lambda.$$

**Case 2:** *When* $x = 0.25$, $y = 1$, *then there exists* $\lambda = 0.25 > 0$ *such that*

$$\Im(J_d(\xi x, \xi y)) - \Im(\mathcal{Q}_\xi^{J_d}(x, y))$$
$$= \Im(J_d(\xi 0.25, \xi 1)) - \Im(\mathcal{Q}_\xi^{J_d}(0.25, 1))$$
$$= \Im(J_d(1, 0.5)) - \Im(\max\{J_d(0.25, 1), J_d(0.25, 1), J_d(1, 0.5), J_d(0.25, 0.5), J_d(1, 1)\})$$
$$= \Im(0.5) - \Im(\max\{0.75, 0.75, 0.5, 0.25, 0\})$$
$$= \Im(0.5) - \Im(0.75) = \ln(0.5) - \ln(0.75)$$
$$= -0.4055 < -0.25 = -\lambda.$$

*This show that* $\Im$ *satisfies* (11) *for all* $x, y \in \mathcal{O}_\xi(x_0)$. *Thus, all hypotheses of Theorem 2.2 hold true and* $\{0, 0.5\}$ *is the set of all fixed points of* $\xi$.

## 4 Consequences

In this section, we find more results involving stronger contractive conditions.

**Corollary 3.1.** *Let* $(z, J_d)$ *be a* $\kappa_{\mathcal{JS}}$*-MS with a preorder* $\sim_{\mathcal{R}}$ *and let* $\xi: z \to z$ *be an* $\sim_{\mathcal{R}}$ *-non-decreasing mapping. Let* $x_0 \in z$ *be a point such that* $x_0 \sim_{\mathcal{R}} \xi x_0$ *and* $\delta_{p_0}(J_d, \xi, x) < \infty$ *for some* $p_0 \in \mathbb{N}$. *Assume there is* $b \in (0, 1)$ *such that*

$$J_d(\xi x, \xi y) \leq b \mathcal{Q}_\xi^{J_d}(x, y), \tag{22}$$

*for all* $x, y \in \mathcal{O}_\xi(x_0)$. *Then the sequence* $\{\xi^p x_0\}_{p \in \mathbb{N}}$ *based on* $x_0$ *is* $\sim_{\mathcal{R}}$ *-non-decreasing and* $J_d$*-Cauchy sequence.*

*Furthermore, if $(z, J_d)$ is $\sim_{\mathcal{R}}$ -non-decreasing-complete, then $\{\xi^p x_0\}_{p \in \mathbb{N}}$ $J_d$-converges to a point $v \in z$ that meets*

$$J_d(v, v) = 0.$$

*Additionally, if $\xi$ is $\sim_{\mathcal{R}}$ -non-decreasing-continuous, then $v = \xi v$.*

*Proof.* Define $\mathfrak{I} : (0, \infty) \to \mathbb{R}$ by $\mathfrak{I}(s) = \ln s$ for all $s \in (0, \infty)$. Put $b = \frac{1}{e^\lambda}$. Inequality (26) implies (11). Hence, all of the requirements of Theorem 2.1 have been met, and the proof is concluded.

**Corollary 3.2**. *Let $(z, J_d)$ be a $\kappa_{\mathcal{JS}}$-MS with a preorder $\sim_{\mathcal{R}}$ and let $\xi: z \to z$ be an $\sim_{\mathcal{R}}$ -non-decreasing mapping. Let $x_0 \in z$ be a point such that $x_0 \sim_{\mathcal{R}} \xi x_0$ and $\delta_{p_0}(J_d, \xi, x) < \infty$ for some $p_0 \in \mathbb{N}$. Assume, there is a non-decreasing function $\mathfrak{I}: (0, \infty) \to (-\infty, \infty)$ fulfilling $(\mathfrak{I}'_2), (\mathfrak{I}_{J_d})$*

$$\xi x \neq \xi y \quad implies \quad \lambda + \mathfrak{I}(J_d(\xi x, \xi y)) \leq \mathfrak{I}(\mathcal{Q}_\xi^{J_d}(x, y)), \quad for \ all \quad x, y \in z \quad such \ that \quad x \sim_{\mathcal{R}} y, \quad (23)$$

*for $\lambda > 0$. Then the sequence $\{\xi^p x_0\}_{p \in \mathbb{N}}$ based on $x_0$ is $\sim_{\mathcal{R}}$ -non-decreasing and $J_d$-Cauchy sequence.*

*Furthermore, if $(z, J_d)$ is $\sim_{\mathcal{R}}$ -non-decreasing-complete, then $\{\xi^p x_0\}_{p \in \mathbb{N}}$ $J_d$-converges to a point $v \in z$ that meets*

$$J_d(v, v) = 0.$$

*Additionally, if $\xi$ is $\sim_{\mathcal{R}}$ -non-decreasing-continuous, then $v = \xi v$.*

*Proof.* Let the contractivity condition (27) hold for all $x, y \in z$ such that $x \sim_{\mathcal{R}} y$, then Lemma 1.1 guarantees that it also holds for $x, y \in \mathcal{O}_\xi(x_0)$. So due to Theorem 2.1, we obtained the result.

**Corollary 3.3**. *Let $(z, J_d)$ be a $\kappa_{\mathcal{JS}}$-MS with a preorder $\sim_{\mathcal{R}}$ and let $\xi: z \to z$ be an $\sim_{\mathcal{R}}$ -non-decreasing mapping. Let $x_0 \in z$ be a point such that $x_0 \sim_{\mathcal{R}} \xi x_0$, $\delta_{p_0}(J_d, \xi, x) < \infty$ for some $p_0 \in \mathbb{N}$ and the following holds true:*

$$\lim_{p_0 + \ell \to \infty} \delta_{p_0 + \ell}(J_d, \xi, x_0) = \lim_{p_0 \to \infty} \delta_{p_0}(J_d, \xi, x_0) \quad for \ arbitrary \quad \ell \in \mathbb{N} \quad such \ that \quad \ell \geq p_0.$$

*If there exists a function $\mathfrak{I} \in \mathcal{W}$ and $\lambda > 0$ fulfilling the inequality (27), then the sequence $\{\xi^p x_0\}_{p \in \mathbb{N}}$ based on $x_0$ is $\sim_{\mathcal{R}}$ -non-decreasing and $J_d$-Cauchy sequence.*

*Furthermore, if $(z, J_d)$ is $\sim_{\mathcal{R}}$ -non-decreasing-complete, then $\{\xi^p x_0\}_{p \in \mathbb{N}}$ $J_d$-converges to a point $v \in z$ that meets*

$$J_d(v, v) = 0.$$

*Additionally, if $\xi$ is $\sim_{\mathcal{R}}$ -non-decreasing-continuous, then $v = \xi v$.*

*Proof.* By using the same reason as in the proof of Corollary 3.2, Theorem 2.2 gives the result.

**Corollary 3.4**. *Let $(z, J_d)$ be a $\kappa_{\mathcal{JS}}$-MS with a partial order $\ll$ and let $\xi: z \to z$ be an $\ll$-non-decreasing mapping. Let $x_0 \in z$ be a point such that $x_0 \ll \xi x_0$ and $\delta_{p_0}(J_d, \xi, x) < \infty$ for some $p_0 \in \mathbb{N}$. Assume, there is a non-decreasing function $\mathfrak{I}: (0, \infty) \to (-\infty, \infty)$ satisfying $(\mathfrak{I}'_2), (\mathfrak{I}_{J_d})$ and (11) for all $x, y \in \mathcal{O}_\xi(x_0)$ and $\lambda > 0$. Then, the sequence $\{\xi^p x_0\}_{p \in \mathbb{N}}$ based on $x_0$ is $\ll$-non-decreasing and $J_d$-Cauchy sequence.*

*Furthermore, if $(z, J_d)$ is $\ll$-non-decreasing-complete, then $\{\xi^p x_0\}_{p \in \mathbb{N}}$ $J_d$-converges to a point $\nu \in z$ that meets*

$$J_d(\nu, \nu) = 0.$$

*Additionally, if $\xi$ is $\ll$-non-decreasing-continuous, then $\nu = \xi \nu$.*

*Proof.* Due to the fact that a partial order $\ll$ is a preorder $\sim_{\mathcal{R}}$, the conclusion is reached by applying Theorem ref 2.1.

**Corollary 3.5.** *Let $(z, J_d)$ be a $\kappa_{\mathcal{JS}}$-MS with a partial order $\ll$ and let $\xi: z \to z$ be an $\ll$-non-decreasing mapping. Let $x_0 \in z$ be a point such that $x_0 \ll \xi x_0$, $\delta_{p_0}(J_d, \xi, x) < \infty$ for some $p_0 \in \mathbb{N}$ and the following holds true:*

$$\lim_{p_0 + \ell \to \infty} \delta_{p_0 + \ell}(J_d, \xi, x_0) = \lim_{p_0 \to \infty} \delta_{p_0}(J_d, \xi, x_0) \quad \text{for arbitrary} \ \ell \in \mathbb{N} \ \text{such that} \ \ell \geq p_0.$$

*If there exists a function $\mathfrak{I} \in \mathcal{W}$ and $\lambda > 0$ fulfilling the inequality (11) for all $x, y \in \mathcal{O}_{\xi}(x_0)$, then the sequence $\{\xi^p x_0\}_{p \in \mathbb{N}}$ based on $x_0$ is $\ll$-non-decreasing and $J_d$-Cauchy sequence.*

*Furthermore, if $(z, J_d)$ is $\ll$-non-decreasing-complete, then $\{\xi^p x_0\}_{p \in \mathbb{N}}$ $J_d$-converges to a point $\nu \in z$ that meets*

$$J_d(\nu, \nu) = 0.$$

*Additionally, if $\xi$ is $\ll$-non-decreasing-continuous, then $\nu = \xi \nu$.*

*Proof.* Due to the fact that a partial order $\ll$ is a preorder $\sim_{\mathcal{R}}$, the conclusion is reached by applying Theorem ref 2.2.

**Corollary 3.6.** *Let $(z, J_d)$ be a $\kappa_{\mathcal{JS}}$-MS with a preorder $\sim_{\mathcal{R}}$ and let $\xi: z \to z$ be an $\sim_{\mathcal{R}}$-non-decreasing mapping. Let $x_0 \in z$ be a point such that $x_0 \sim_{\mathcal{R}} \xi x_0$ and $\delta_{p_0}(J_d, \xi, x) < \infty$ for some $p_0 \in \mathbb{N}$. Assume, there is a non-decreasing function $\mathfrak{I}: (0, \infty) \to (-\infty, \infty)$ satisfying $(\mathfrak{I}'_2)$, $(\mathfrak{I}_{J_d})$ and*

$$\lambda + \mathfrak{I}(J_d(\xi x, \xi y)) \leq \mathfrak{I}\left( \max\{J_d(x, y), \frac{J_d(x, \xi x) + J_d(y, \xi y)}{2}, \frac{J_d(x, \xi y) + J_d(y, \xi x)}{2}\}\right), \quad (24)$$

*for all $x, y \in \mathcal{O}_{\xi}(x_0)$ with $\xi x \neq \xi y$ and $\lambda > 0$. Then the sequence $\{\xi^p x_0\}_{p \in \mathbb{N}}$ based on $x_0$ is $\sim_{\mathcal{R}}$-non-decreasing and $J_d$-Cauchy sequence.*

*Furthermore, if $(z, J_d)$ is $\sim_{\mathcal{R}}$-non-decreasing-complete, then $\{\xi^p x_0\}_{p \in \mathbb{N}}$ $J_d$-converges to a point $\nu \in z$ that satisfies*

$$J_d(\nu, \nu) = 0.$$

*Additionally, if $\xi$ is $\sim_{\mathcal{R}}$-non-decreasing-continuous, then $\nu = \xi \nu$.*

*Proof.* Since $\frac{(r+s)}{2} \leq \max\{s, r\}$ for all s, r $\in [0, \infty]$, so inequality (28) implies inequality (11) and Theorem 2.1 leads to the conclusion.

**Corollary 3.7.** *Let $(z, J_d)$ be a $\kappa_{\mathcal{JS}}$-MS with a preorder $\sim_{\mathcal{R}}$ and let $\xi: z \to z$ be an $\sim_{\mathcal{R}}$-non-decreasing mapping. Let $x_0 \in z$ be a point such that $x_0 \sim_{\mathcal{R}} \xi x_0$ and $\delta_{p_0}(J_d, \xi, x) < \infty$ for some $p_0 \in \mathbb{N}$. Assume, there is a continuous non-decreasing function $\mathfrak{I}: (0, \infty) \to (-\infty, \infty)$ obeying*

$$\lambda + \mathfrak{I}(J_d(\xi x, \xi y)) \leq \mathfrak{I}\left( \max\{J_d(x, y), J_d(x, \xi x), J_d(y, \xi y), \frac{J_d(x, \xi y) + J_d(y, \xi x)}{2}\}\right), \quad (25)$$

*for all* $x, y \in \mathcal{O}_\xi(x_0)$ *with* $\xi x \neq \xi y$, $\lambda > 0$ *and the following*:

$$\lim_{p_0 + \ell \to \infty} \delta_{p_0 + \ell}(J_d, \xi, x_0) = \lim_{p_0 \to \infty} \delta_{p_0}(J_d, \xi, x_0) \quad \textit{for arbitrary} \quad \ell \in \mathbb{N} \quad \textit{such that} \quad \ell \geq p_0.$$

*Then the sequence* $\{\xi^p x_0\}_{p \in \mathbb{N}}$ *based on* $x_0$ *is* $\sim_\mathcal{R}$ *-non-decreasing and* $J_d$*-Cauchy sequence.*

*Furthermore, if* $(z, J_d)$ *is* $\sim_\mathcal{R}$ *-non-decreasing-complete, then* $\{\xi^p x_0\}_{p \in \mathbb{N}}$ $J_d$*-converges to a point* $\nu \in z$ *that obeys*

$$J_d(\nu, \nu) = 0.$$

*Additionally, if* $\xi$ *is* $\sim_\mathcal{R}$ *-non-decreasing-continuous, then* $\nu = \xi \nu$.

*Proof.* Since $\frac{(r+s)}{2} \leq \max\{s, r\}$ for all $s, r \in [0, \infty]$, so inequality (28) implies inequality (11) and the result follows from Theorem 2.2.

**Remark 3.1**. *Theorem 2.1, Theorem 2.2 and Corollaries 3.1-3.7 remain true if we do one or more of the following changes in their statement*:

- *exchange the* $\sim_\mathcal{R}$*-non-decreasing-completeness of* $(z, J_d)$ *by the completeness of* $(z, J_d)$;

- *exchange the* $\sim_\mathcal{R}$*-non-decreasing-continuity of* $\xi$ *by continuity*;

- *exchange the preorder* $\sim_\mathcal{R}$ *by the trivial preorder* $\sim_{\mathcal{R}z}$ *given by* $x \sim_{\mathcal{R}z} y$ *for all* $x, y \in z$;

- *exchange, in the contractivity condition, for all* $x, y \in \mathcal{O}_\xi(x_0)$ *by for all* $x, y \in z$ *such that* $x \sim_\mathcal{R} y$;

- *exchange the* $\kappa_{\mathcal{JS}}$*-MS by any of the abstract metric spaces that could be considered as a* $\kappa_{\mathcal{JS}}$*-MS: b-metric spaces, modular spaces, and standard metric spaces.*

- *exchange the contractivity condition* (11) *by*

$$\lambda + \mathfrak{I}(J_d(\xi x, \xi y)) \leq \mathfrak{I}(J_d(x, y)),$$

*for every* $x, y \in \mathcal{O}_\xi(x_0)$ *with* $\xi x \neq \xi y$.

## 5 Existence of solution to RLC circuit's current differential equation

A tuning circuit is a mathematical representation of the electric current in an RLC parallel circuit to present with a rudimentary knowledge of how light is converted into electricity. Consider the following series of electric circuit (Fig 1), which includes a resistor R, a capacitor C, an inductor L, a voltage V, and an electromotive force E. With the aid of Kirchhoff's Voltage Law, related problems are mathematically modelled as initial value problems for second order ordinary differential equations of the form:

$$\begin{cases} L\dfrac{d^2 q}{dt^2} + R\dfrac{dq}{dt} + \dfrac{q}{C} = V_\nu(t) \\ q(0) = q'(0) = 0, \end{cases} \tag{26}$$

where $V_\nu(t) = V$.

In this part, we demonstrate the existence of the solution to the RLC differential equation (26). The problem (26) is identical to the following integral equation (see [29, 30]:

$$q(t) = \int_0^t \mathcal{G}(t, g)\hbar(g, q(g))dg, \quad t \in [0, 1], \tag{27}$$

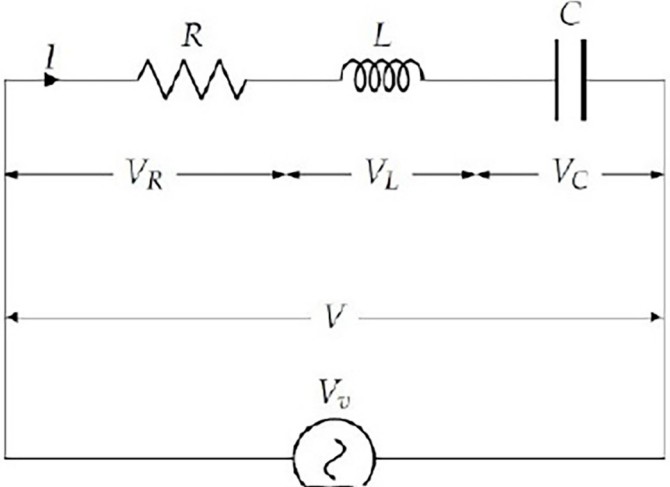

FIGURE1.RLC-SeriesCircuit

**Fig 1. RLC series circuit.**

where $\hbar : [0, 1] \times \mathbb{R} \to \mathbb{R}$ is a monotonically non-decreasing function for all $g \in [0, 1]$ and $\mathcal{G}$ is the Green function defined as

$$\mathcal{G}(t, g) = \begin{cases} -ge^{-\tau(g-t)} & \text{if} \quad 0 \leq g \leq t \leq 1; \\ -te^{-\tau(g-t)} & \text{if} \quad 0 \leq t \leq g \leq 1, \end{cases} \tag{28}$$

$\tau$ is a constant computed in terms of R and L. Let $z = C(J, \mathbb{R})$ be the space of all continuous real valued functions on J, where J = [0, 1]. Then $z$ is a complete metric space with respect to the metric $J_d(\hat{a}, \hat{b}) = \|\hat{a} - \hat{b}\|_\infty = \sup_{t \in J} |\hat{a}(t) - \hat{b}(t)|$ and so $z$ is $\kappa_{\mathcal{JS}}$-MS for $\kappa = 1$. Hereafter, we assume that $(z, J_d)$ is a $\kappa_{\mathcal{JS}}$-MS with canonical preorder $\leq$ and $(z, J_d)$ is $\leq$-non-decreasing-complete. Define the operator $\aleph: z \to z$ as follows:

$$\aleph(q(t)) = \int_0^t \mathcal{G}(t, g)\hbar(g, q(g))dg, \quad t \in [0, 1]. \tag{29}$$

A fixed point of operator (29) is the solution of problem (26). We take into account the following hypotheses:

(H1) $\mathcal{G}: J^2 \to [0, \infty)$ is a continuous function;

(H2) $|\hbar(t, q(t)) - \hbar(t, p(t))| \leq |q(t) - p(t)| + 1$ for all $t \in [0, 1]$;

(H3) $|\mathcal{G}|_\infty = \sup_{t \in I} \int_0^t \mathcal{G}(t, g) \leq 1$;

(H4) $\aleph$ is $\leq$-non-decreasing continuous.

**THEOREM 4.1**. *Suppose that hypothesis* (H1)-(H4) *hold. Then, the initial value problem* (26) *has a common solution in z.*

*Proof.* Firstly, note that for $\Im(\hat{a}) = -\frac{1}{\hat{a}+1}$ for all $\hat{a} \in (0, \infty)$, inequality (11) is equivalent to the following for all x, y $\in z$:

$$J_d(\xi x, \xi y) \leq \frac{\mathcal{Q}_\xi^{J_d}(x,y)+1}{1+\lambda \mathcal{Q}_\xi^{J_d}(x,y)} - 1 \quad \leq \frac{\mathcal{Q}_\xi^{J_d}(x,y)+1}{1+\lambda \mathcal{Q}_\xi^{J_d}(x,y)} \leq \mathcal{Q}_\xi^{J_d}(x,y)+1. \tag{30}$$

Next, for all $p, q \in \mathcal{O}_\aleph(p_0)$ and t $\in$ J, we have

$$
\begin{aligned}
|\aleph(q(t)) - \aleph(p(t))| &= \left| \int_0^t \mathcal{G}(t,g)[\hbar(g, q(g)) - \hbar(g, p(g))]dg \right| \\
&\leq \int_0^t \mathcal{G}(t,g)|[\hbar(g, q(g)) - \hbar(g, p(g))]|dg \\
&\leq \int_0^t \mathcal{G}(t,g)|q(t) - p(t) + 1|dg \\
&\leq \int_0^t \mathcal{G}(t,g)(\max\{|q(t) - p(t)|, |\aleph(p(t)) - p(t)|, |\aleph(q(t)) - q(t)|, \\
&\quad |\aleph(q(t)) - p(t)|, |\aleph(p(t)) - q(t)|\} + 1)dg,
\end{aligned}
$$

This implies that

$$
\|\aleph(q) - \aleph(p)\|_\infty \leq |\mathcal{G}|_\infty (\max\{\|p - q\|_\infty, \|\aleph(p) - p\|_\infty, \|\aleph(q) - q\|_\infty, \|\aleph(q) - p\|_\infty, \\
\|\aleph(p) - q\|_\infty\} + 1). \tag{31}
$$

From (H3) and (31), we have

$$\|\aleph(q) - \aleph(p)\|_\infty \leq \max\{\|p - q\|_\infty, \|\aleph(p) - p\|_\infty, \|\aleph(q) - q\|_\infty, \|\aleph(q) - p\|_\infty, \|\aleph(p) - q\|_\infty\} + 1.$$

Hence (11) is satisfied for $\Im(\hat{a}) = -\frac{1}{\hat{a}+1}$. Thus, all hypotheses of Theorem 2.2 are satisfied and therefore differential Eq (26) has a solution in J.

## 6 Conclusion

In this work, we establish $\Im$-contractions and show some fixed point theorems for these contractive conditions in the JS-generalized metric spaces. Finally, we proved fixed point results, an existence result for the solution of the RLC circuit's current differential equation is also established.

## Acknowledgments

Authors are thankful to the editor and anonymous referees for their valuable comments and suggestions.

## Author Contributions

**Conceptualization:** Bilal Iqbal, Naeem Saleem, Asima Razzaque.

**Investigation:** Maggie Aphane.

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
