## [Decision Letter · Decision Letter 0]

5 Aug 2024

PONE-D-24-27988Fixed Point Results for Weak F-Contractions in JS Generalized Metric Spaces with an ApplicationPLOS ONE

Dear Dr. Saleem,

Thank you for submitting your manuscript to PLOS ONE. After careful consideration, we feel that it has merit but does not fully meet PLOS ONE’s publication criteria as it currently stands. Therefore, we invite you to submit a revised version of the manuscript that addresses the points raised during the review process.

Please submit your revised manuscript by Sep 19 2024 11:59PM. If you will need more time than this to complete your revisions, please reply to this message or contact the journal office at plosone@plos.org. Please include the following items when submitting your revised manuscript:A rebuttal letter that responds to each point raised by the academic editor and reviewer(s). You should upload this letter as a separate file labeled 'Response to Reviewers'.A marked-up copy of your manuscript that highlights changes made to the original version. You should upload this as a separate file labeled 'Revised Manuscript with Track Changes'.An unmarked version of your revised paper without tracked changes. You should upload this as a separate file labeled 'Manuscript'.If applicable, we recommend that you deposit your laboratory protocols in protocols.io to enhance the reproducibility of your results. Protocols.io assigns your protocol its own identifier (DOI) so that it can be cited independently in the future. For instructions see: https://journals.plos.org/plosone/s/submission-guidelines#loc-laboratory-protocols. Additionally, PLOS ONE offers an option for publishing peer-reviewed Lab Protocol articles, which describe protocols hosted on protocols.io. Read more information on sharing protocols at https://plos.org/protocols?utm_medium=editorial-email&utm_source=authorletters&utm_campaign=protocols.

We look forward to receiving your revised manuscript.

Kind regards,

Rizwan Anjum

Academic Editor

PLOS ONE

Journal Requirements:

3. Please ensure that you refer to Figure 1 in your text as, if accepted, production will need this reference to link the reader to the figure.

4. Please include a caption for figure 1. 

Reviewers' comments:

Reviewer's Responses to Questions

**Comments to the Author**

1. Is the manuscript technically sound, and do the data support the conclusions?

Reviewer #1: Yes

Reviewer #2: Yes

2. Has the statistical analysis been performed appropriately and rigorously? 

Reviewer #1: Yes

Reviewer #2: No

3. Have the authors made all data underlying the findings in their manuscript fully available?

Reviewer #1: Yes

Reviewer #2: Yes

4. Is the manuscript presented in an intelligible fashion and written in standard English?

Reviewer #1: Yes

Reviewer #2: Yes

5. Review Comments to the Author

Reviewer #1: I have gone through the article; the manuscript consists of fixed-point results for weak contractions. Finally, using proven fixed point results, the existence of the solution of the RLC circuit's current differential equation is proved.

The results in this paper are intriguing, informative, and accurate. The article is sound and deserves to be published in Plos One. it will contribute positively to the literature, and the novelty of the results is high.

However, some minor changes are required as mentioned below:

1. On page 2, replace “in numerous ways. (see [4, 13, 18, 19, 20, 21, 24, 28])” with “in numerous ways (see [4, 13, 18, 19, 20, 21, 24, 28]).”

2. Add a potential example of the theoretical results (Main Result) for the understanding of the readers and to show the applicability.

3. At page 2, in 3rd paragraph, add space in the following: “ in[27].”, “Hereafter”.

4. In Remark 1.3, replace “complete As” with “complete. As”

5. There are several other typo mistakes in the article, authors are encouraged to revise the article as per suggestions.

Reviewer #2: Dear Author

Be careful on the text of your paper. See my comments which I highlighted them in yellow.

Reply all of them separately. I will read again your paper. Your idea is good generally. But in application part you should consider suitble control functions. Not that you work on single valued mappings, so we have x=f(x) for a fixed point.

Best

6. PLOS authors have the option to publish the peer review history of their article (what does this mean?). If published, this will include your full peer review and any attached files.

Reviewer #1: No

Reviewer #2: No

---

## [Author Response · Author response to Decision Letter 0]

22 Oct 2024

Dear Editor,

Plos One

I am pleased to re-submit our paper entitled \\textquotedblleft

Fixed Point Results for Weak $\\fs$-Contractions in JS-Generalized Metric Spaces with an Application\\textquotedblright ,\\ (with A. Razzaque, B. Iqbal, and M. Aphane ) in your

respective journal Plos One.

We would like to express our sincere gratitude for your editorial efforts in completing the first round of reviews. We are also thankful to the reviewers for their insightful comments and suggestions, which have significantly contributed to improving our manuscript.

After carefully considering the reviewers' feedback, we have made substantial revisions to enhance the readability and overall quality of the article. We have addressed each comment and suggestion provided by the reviewers, and have made corresponding improvements to the manuscript.

We have attached the revised version of the article, along with a highlighted PDF that includes our responses to all the reviewers' comments. The revised version has also been uploaded to the system.

Thank you for your message. I confirm that all the necessary information and materials required to replicate the study in its entirety have been provided in the submission files. Should you require any further clarification or additional details, please do not hesitate to reach out.

I appreciate your attention to this matter and look forward to moving forward.

We kindly request that you proceed with the next steps in the editorial process.

Thank you for your continued support. 

Best Regards,

Naeem Saleem

---

## [Editor Report · Decision Letter 1]

12 Nov 2024

Fixed Point Results for Weak F-Contractions in JS Generalized Metric Spaces with an Application

PONE-D-24-27988R1

Dear Dr. Saleem,

We’re pleased to inform you that your manuscript has been judged scientifically suitable for publication and will be formally accepted for publication once it meets all outstanding technical requirements.

Kind regards,

Rizwan Anjum

Academic Editor

PLOS ONE
---

## [Editor Report · Acceptance letter]

19 Nov 2024

PONE-D-24-27988R1 

PLOS ONE

Dear Dr. Saleem, 

I'm pleased to inform you that your manuscript has been deemed suitable for publication in PLOS ONE. Congratulations! Your manuscript is now being handed over to our production team.

Kind regards, 

on behalf of

Dr. Rizwan Anjum 

Academic Editor

PLOS ONE